# Transcriptome sequencing reveals the evolutionary histories and gene expression evolution in two related *Pagurus* species

Zakea Sultana[1], Isaac Adeyemi Babarinde[2]*, Masafumi Nozawa[3,4], Kazuho Ikeo[5,6], Akira Asakura[7], Tomoyuki Nakano[7]*

1 Fisheries Division, Japan International Research Center for Agricultural Sciences, Tsukuba, Ibaraki, Japan, 2 Laboratory of Inflammation and Vaccines (LIV), Shenzhen Institutes of Advanced Technology, Chinese Academy of Sciences, Shenzhen, Guangdong Province, China, 3 Department of Biological Sciences, Tokyo Metropolitan University, Tokyo, Japan, 4 Research Center for Genomics and Bioinformatics, Tokyo Metropolitan University, Tokyo, Japan, 5 Laboratory for DNA Data Analysis, National Institute of Genetics, Mishima, Shizuoka, Japan, 6 Department of Genetics, School of Life Science, The Graduate University for Advanced Studies (SOKENDAI), Mishima, Shizuoka, Japan, 7 Seto Marine Biological Laboratory, Field Science Education and Research Center, Kyoto University, Nishimuro, Wakayama, Japan

* adeyemi@siat.ac.cn (IAB); nakano.tomoyuki.2a@kyoto-u.ac.jp (TN)

## Abstract

*Pagurus lanuginosus* (De Haan, 1833–1850) and *Pagurus maculosus* (Komai and Imafuku, 1996) are two closely related species of the genus *Pagurus*, generally referred to as "right-handed hermit crabs". Previously thought to be color morphs of the same species, recent studies have itemized their unique features. To investigate the molecular and gene expression evolution that have followed the divergence of these two species, we performed transcriptome sequencing and *de novo* transcriptome assembly using cephalothorax (head) and pereopod (leg) of the two species. Synonymous and nonsynonymous divergence between the two species was estimated to be 0.0246 and 0.0066, respectively. About 5% of the protein-coding transcripts had signatures of positive selection. The phylogenetic analyses showed that the two species are indeed closely related and diverged about 6.24 million years ago. Comparison of tissue expression showed that head expressions were more conserved, and that more genes were upregulated in the legs of *P. maculosus* leg; while more pigmentation genes were found to be upregulated in the legs of *P. lanuginosus*. Genes associated with the extracellular matrix and space, chitin binding and chitin-based larval cuticles were found to have higher expression in the legs of *P. maculosus*, suggesting the roles in morphological differences. This study sheds light on the evolutionary history of the two species at the molecular level.

**Data availability statement:** All data generated in the study have been made publicly available and deposited to the DNA Data Bank of Japan under the project accession number PRJDB35803, and can be accessed on the NCBI database (https://www.ncbi.nlm.nih.gov/bioproject/PRJDB35803/). All short-read data have been described in S1 Table.

**Funding:** This study was supported by the funding of Research Organization of Information and Systems (ROIS) for special collaboration research student program of National Institute of Genetics (NIG) and Kyoto University, Japan awarded to ZS, and by the Operating Expenses (運営費) from Kyoto University, Japan awarded to AA and TN. The funders had no role in study design, data collection and analysis, decision to publish, or preparation of the manuscript.

**Competing interests:** The authors have declared that no competing interests exist.

## Introduction

The genus *Pagurus* [1], generally referred to as 'right-handed hermit crabs', is widely distributed in almost all marine habitats ranging from shallow to deep waters, worldwide. The genus is found in diverse biotopes along hard rocky shores to intertidal muddy flats. Thus, the species of this genus exhibit extremely high morphological diversity [2]. The morphological diversity and the ecological heterogeneity associated with the species of this genus make it difficult to identify and classify *Pagurus* species. Consequently, many sibling species remain cryptic species that are difficult to recognize using classic systematic methods [3]. Many new species are being described and added to this genus continuously, even as many more are yet to be recognized [4]. Therefore, the taxonomic classification of this ever-proliferative genus *Pagurus* is far from being resolved [5].

This concern is exemplified in two Japanese water intertidal hermit crab species, *Pagurus lanuginosus* [6] and *P. maculosus* [7]. Because of their superficial similarities, the two species were previously considered to be two color morphs of one species, *P. lanuginosus* [8]. The major difference was the color of their chromatophores scattered on the pereopods (legs) and cephalothorax (head). The chromatophores are white in *P. maculosus* and black in *P. lanuginosus* [7,8]. They both inhabit intertidal rocky shores and sometimes exhibit cohabitation [9]. However, the distribution of *P. lanuginosus* is broader than that of *P. maculosus*. Additional differences in morphological characters such as the number of calcareous teeth on the dactyl and the presence or absence of a large tubercle on the merus of the right cheliped in adults were described by Komai and Imafuku [7]. Thus, Komai and Imafuku concluded that these hermit crabs are two different species [7]. Further, careful studies of the larval and post-larval development of these two species revealed differences in the number of setae or spines on the antennule, maxillule and maxilla, and the presence of chromatophores on the carapace [10,11].

Morphological characters have been classically used for the identification and descriptions of decapod crustaceans. However, in some cases, morphological characters were very similar between congeners, raising uncertainty about their independent species status. With the advancement in nucleic acid extraction [12] and sequencing technologies [13], recent attention has been focused on the evolutionary studies of diverse decapod groups, utilizing the molecular approaches to disentangle various unresolved questions on the origins of diversity, habitat shifts, and diversification [14–17]. Molecular evolution has now been used as the basis for investigating morphological evolution [18–20].

Previous molecular evolution attempts using mitochondrial loci showed that *P. maculosus* and *P. lanuginosus* were indeed phylogenetically very close [9,21]. In this study, we aimed to investigate the phylogenetic relationships of the two species with other related species with additional focus on the gene expression evolution using transcriptome data. To achieve this, we generated head and leg transcriptomes of the two species. We conducted *de novo* transcript assembly and functionally annotated the newly assembled transcripts. We used the assembled transcripts to reveal some insights into the molecular evolution, phylogenetic relationship and gene expression evolution in the two hermit crab species.

## Materials and methods

### Sampling location and permit

Specimens of *Pagurus lanuginosus* and *P. maculosus* were collected from the same location at a rocky shore at Shirahama (33°41' N, 135°23' E), Wakayama Prefecture, along the North Pacific, Japan, in November 2023, and were transported to the laboratory alive. Sample collection permit for experimental research with number 5−2 was granted by Wakayama Prefecture, Japan. Researches on *Pagurus* species do not require any special approval. All experiments complied with institutional regulations and Japanese policy on animal use [22].

### RNA extraction and sequencing

Three individuals of each species (*P. maculosus* and *P. lanuginosus*) were sampled for RNA extraction. For each individual, RNA was extracted from two distinct tissues in which pigmentation is most prevalent: the cephalothorax (head) and pereopod (leg), as illustrated in Fig 1A and Fig 1B. This resulted in three biological replicates per tissue type for each species. Total RNA was extracted by homogenizing approximately 25 mg of head or leg in 500 µl of Binding Buffer (4 M Guanidine Thiocyanate; 25 mM Na Citrate; 0.5% SDSC; 0.1 M Mercaptoethanol) with a tissue homogenizer (BEADS CRUSHER µT-12, TAITEC). The homogenate was processed following the manufacturer's protocol of PureLink RNA Mini Kits (Qiagen). RNA purity and integrity were assessed visually in agarose gels, and quantified in spectrophotometry (Nanodrop ND-1000, NanoDrop Technologies, USA). High-quality total RNA samples were sent to an outside facility (Macrogen, Japan) for subsequent steps including library preparation and next generations sequencing. Library preparation was done using TruSeq Stranded Total RNA Library Preparation Kit (Illumina) and the sequencing was done on the NovaSeq 6000 system (Illumina) platform with 150 base pair paired-end (PE) reads. Head and leg of each species were sequenced in triplicates, with 10Gb of data per sample. All raw reads generated in this study are available in the DDBJ Sequence Read Archive (DRA) under the BioProject accession number PRJDB35803. The data sets supporting the results presented here are available with the accession numbers DRR708008-DRR708019.

### Retrieval of publicly available annotation data and crustacean short-read data

Publicly available data were retrieved for annotation and evolutionary analyses. Nonredundant (nr) amino acid sequences were retrieved from the NCBI database. The crustacean species with the highest homology were selected. The short-read raw RNA-seq data of the top homologous crustacean species, with at least 20,000 homologous transcripts, were then retrieved from the NCBI short-read archives. In addition, we retrieved the short-read RNA-seq data from the project PRJNA562428 [23]. Both newly generated RNA-seq data and publicly available data were then independently assembled using the same *de novo* transcript assembly pipeline.

### Transcriptome assembly and its evaluation

The quality of the sequenced reads was examined using version v0.11.9 of fastQC (https://www.bioinformatics.babraham.ac.uk/projects/fastqc). For transcript assembly, raw reads of all tissues and replicates of the same species were concatenated. *De novo* transcript assembly was performed using the default settings of the version v2.13.2 of Trinity [24,25]. Initial transcript expression quantification was done using bowtie2 in RSEM [26], with the settings "*--bowtie2 --paired-end --bowtie2-sensitivity-level very_sensitive*". Transcripts with less than 2 RSEM's expected counts were discarded. To evaluate the quality of the assembly, bowtie2 [27] was used to map the raw reads back to the assembled transcripts to quantify percentage of reads that were used for the actual assembly. The filtered transcript sets were then used for the downstream analyses.

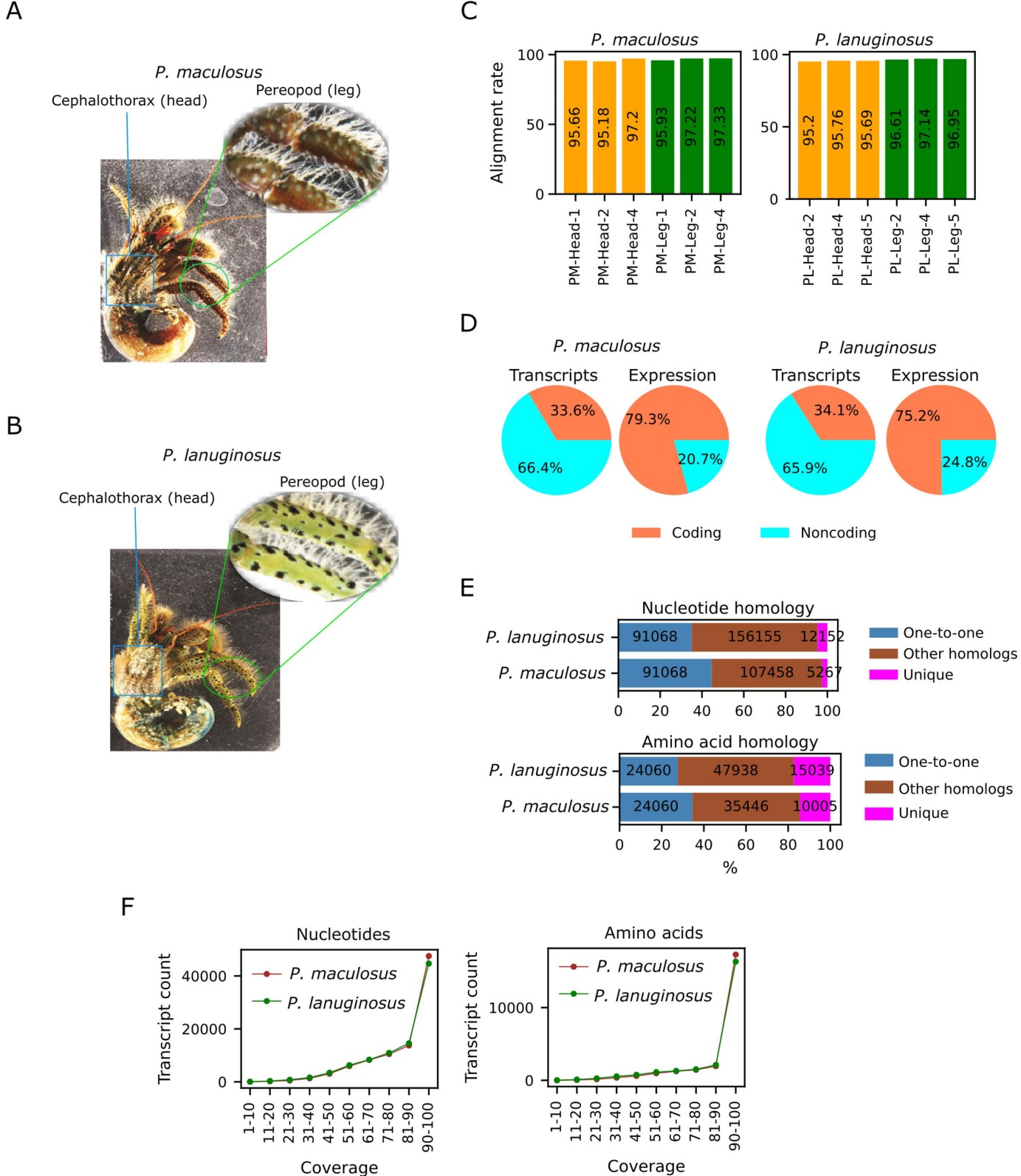

**Fig 1. Transcriptome sequencing of *Pagurus maculosus* and *P. lanuginosus*.** A. The picture of *P. maculosus* showing the parts from which RNA was extracted. B. The picture of *P. lanuginosus* showing the parts from which RNA was extracted. C. Alignment rates of reads that could be mapped

back to the assembled transcripts. D. The distribution of coding and noncoding transcripts and expression. The transcript pie charts show the distribution of the transcript counts while the expression pie charts show contributions to the expression levels. E. Homology of a species in the other species. Unique transcripts did not have significant similarity with any transcripts in the other species. Transcripts with homology were divided into "one-to-one" or "other homologs" based on one-to-one correspondence in the two species. F. Coverage of one-to-one transcripts in the other species.

## Transcript expression quantification

Transcript expression quantification was performed using RSEM with bowtie2 alignment option. First, RSEM reference was built for the transcript assembly of each species. The raw reads were cleaned with fastp [28] with the settings "*-q 20 -u 10*". The fastp-filtered reads were then used for the expression quantification. The RSEM settings used were "*--bowtie2 --paired-end --bowtie2-sensitivity-level very_sensitive*".

## Transcript annotation

Transcript annotation was focused on the predicted protein-coding transcripts. First, the TransDecoder [25] was used to extract all the six open reading frames (ORFs) with at least 50 amino acids. The most likely protein-coding ORF was then predicted by TransDecoder in Trinity Package. TransDecoder-predicted amino acid sequences were then used as queries for homology search to identify the homologous annotated proteins in NCBI Reference Sequence (RefSeq) non-redundant (nr) [29,30], Uniprot reference (UniRef90) [31] and UniProtKB/TrEMBL [32] databases. Homology search was done using DIAMOND [33] with "*--sensitive*" mode.

Functional annotation of the predicted amino acids was done using InterProScan 5 [34]. The predicted gene ontology (GO) terms and pathways were also included in the output. For the annotation of pigmentation-related genes, we employed a published strategy [35]. Briefly, the term "pigmentation" was searched in the gene ontology database [36] to obtain the list of genes with pigmentation functions in any species. The sequences of the genes were then obtained from UniProtKB/TrEMBL database to obtain the pigmentation gene reference sequences. Predicted amino acids were then searched in the pigmentation reference using BLASTP [37] with "*--evalue 0.001*" to obtain the list of pigment-related crab genes.

## Gene ontology enrichment analyses

Gene ontology (GO) enrichment analyses were performed using the results of InterProScan. Genes with no annotated GO term was not included in the enrichment analyses. Similarly, GO terms with no annotated gene were excluded. GO analyses were computed for positively selected gene set and differentially expressed genes. For each GO term, the enrichment and hypergeometric probability were computed. Multiple testing was corrected with Bonferroni procedure [38]. Adjusted p value$<0.05$ was set as the threshold for significantly enriched GO term.

## Evolutionary analyses

Evolutionary analyses were conducted at both the nucleotide and amino acid levels. Nucleotide homology search was done with BLASTN [37] while amino acid searches were done with DIAMOND or BLASTP, depending on the size of the database. Reciprocal best hits, assessed by the bit scores, between the two species were termed "one-to-one". The identification of the coding sequences (CDS) was done with TransDecoder. Stop codons were striped from the CDS. Each cluster of one-to-one genes were aligned with CLUSTALW [39]. Custom python scripts were used to concatenate the sequences, and partition them into different codon positions. The best amino acid and nucleotide models were estimated using maximum likelihood models in MEGA [40]. The rates of synonymous (dS) and nonsynonymous (dN) substitutions using both Nie-Gojobori [41] and Yang-Nielsen [42] methods were computed with *yn00* in version 4.10.6 of PAML [43]. The evolutionary pressures acting on the transcripts were estimated using the estimated and error values of dS and dN

computed from Yang-Nielsen method [42]. For positively selected transcripts, $dN-dN_{err} > dS + dS_{err}$, where $dN_{err}$ and $dS_{err}$ are error estimates for dN and dS, respectively. For purifying selection, $dS-dS_{err} > dN + dN_{err}$. Other transcripts were classified as neutrally evolving.

### Phylogenetics and divergence time estimates

Because of the computational requirements of the maximum likelihood, only one fourth of the amino acid positions of the concatenated amino acids were used for the phylogenetic tree. To extract the alignment subset, the amino acids of the first position in 4-step moving window were extracted using a custom Python script. The phylogenetic analyses were conducted using maximum likelihood method in MEGA. Aligned sites with gaps or unknown amino acids were completely deleted. Gamma distribution with invariant sites (G + I) and gamma = 5, was used for the rate among sites. The initial tree was made with NJ/BioNJ method. Weak branch swap filter was set. LG model with frequencies were used. Branch support was evaluated with 1,000 bootstrap replicates.

Divergence time estimates were computed using MCMCtree program available in version 4.10.6 of PAML following previously published methods [44–46]. First, the overall substitution rates were estimated using *baseml* in PAML with the root age set to 450 million years ago (MYA). Next, the gradient (g) and Hessian (H) were estimated using the phylogenetic tree and alignments of the multiple species. Finally, the divergence times were estimated using the computed gradient, Hessian, alignments and the fossil calibrations. Nine calibration points were retrieved from published studies [17,47–49]. To assess the convergence, the divergence time estimation was repeated three times with different random seed numbers.

### Differential gene expression

The consistencies of the replicates were first examined by principal component analyses and clustering analyses using the expression data normalized by DESeq2 [50]. The two most consistent samples for each species-tissue combination were used for differential gene expression analyses. Differential gene expressions were computed for each tissue between the two species using DESeq2. Differential gene expression was done independently for head and leg. Statistical significance was set at adjusted p value of 0.05.

### Statistical tests and graphical representations

Statistical tests were computed using Scipy package in Python or R statistical package. For gene enrichment analyses, hypergeometric tests were conducted to test the significance of the enrichment of specific gene ontology and term. Multiple tests were corrected using Bonferroni method [38]. Graphical representations were done mostly by Matplotlib in Python and occasionally by R.

## Results

### Head and leg transcriptome assemblies of *P. maculosus* and *P. lanuginosus*

RNA sequencing was performed with three biological replicates from each tissue type and species. For each replicate, ~21,000,000–27,000,000 paired-end reads were sequenced (S1 Table). A total of 139,881,387 *P. lanuginosus* and 148,561,226 *P. maculosus* paired-end reads were generated (Table 1). Transcriptome assembly produced 373,660 and 468,052 initial transcripts for *P. maculosus* and *P. lanuginosus,* respectively. Deeper sequencing has been reported to be positively correlated with the number of assembled transcripts, including potentially noisy transcripts [51]. Removal of potentially noisy transcripts produced 203,793 transcripts, including 140,954 genes for *P. maculosus,* and 259,375 transcripts, including 179,685 genes for *P. lanuginosus* (Table 1). More than 95% of raw reads could be mapped back to the filtered assembled transcripts (Fig 1C), showing that the transcriptomes were sufficiently represented in the transcript assemblies.

**Table 1. *De novo* transcript assembly statistics.**

| Metrics | *Pagurus lanuginosus* | *Pagurus maculosus* |
|---|---|---|
| Total raw read count | 139881387 | 148561226 |
| Assembled gene count | 179685 | 140954 |
| Assembled transcript count | 259375 | 203793 |
| Transcript N50 | 1029 | 1084 |
| Transcript median length (bp) | 470 | 498 |
| Transcript average length (bp) | 765 | 804 |
| Predicted coding transcripts* | 87037 | 69511 |

*Transcripts with open reading frame (orf) with at least 50-amino acid were predicted to be protein-coding.

Transcriptome assembly has the potential to retrieve both coding and noncoding transcripts. We therefore checked the proportion of both coding and noncoding transcripts in our transcriptome assembly (see Materials and Methods). In both species, about 34% of the transcripts were coding (Fig 1D). However, expression quantifications showed that coding transcripts constituted about 79% and 75% of the *P. maculosus* and *P. lanuginosus* transcriptomes, respectively, reflecting higher expression levels of coding transcripts, as previously reported in human [52,53].

Next, we checked the homology between the assembled *P. maculosus* and *P. lanuginosus* transcripts. At the nucleotide levels, >95% of *P. lanuginosus* transcripts had homologs in *P. maculosus* (Fig 1E). On the other hand, >97% *P. maculosus* transcripts had homologs in *P. lanuginosus.* At the amino acid levels, 83% of the *P. lanuginosus* transcripts and 86% of *P. maculosus* transcripts had detectable homologs in the other species, at the specified search thresholds. We defined one-to-one homology as the reciprocal best hit in the two species in reciprocal BLAST searches. Interestingly, although the majority of the transcripts had homologs in the two species, one-to-one relationship could not be established for many transcripts. The inability to establish one-to-one relationship could suggest the presence of transcriptional noise, false positives or false negatives reflecting the limitation of transcript assembly and annotation procedures from short-read RNA-seq [51,53]. This observation could also highlight the roles of independent evolution of gene duplications, losses and/or splicing events after the divergence of the two species. For transcripts with one-to-one correspondence, a substantial proportion of the transcripts had 90–100% alignment coverage in the two species (Fig 1F), highlighting the confidence of our assembly and annotation, at least for transcripts with one-to-one relationship. These results highlight the properties of the assembled transcripts and the evolution of gene expressions after the divergence of the two species.

## Annotation of the assembled *Pagurus* transcripts

To functionally annotate the assembled transcripts, we searched for the predicted amino acids in NCBI RefSeq nr, UniRef90 and UniProtKB TrEMBL amino acid databases. We defined homology by significant similarity with the annotated proteins. About 48,000 *P. lanuginosus* and about 39,000 *P. maculosus* transcripts had homology in the databases (S2 Table, upper panel of Fig 2A). Interestingly, 48,712 *P. lanuginosus* transcripts had homolog in at least one of the investigated databases, including 47,653 (97.8%) with homology in the three investigated databases (S2 Table). Similarly, 39,639 *P. maculosus* transcripts had homolog in at least one of the three databases, 38,805 (97.9%) of which were found in all three databases. The consistency across databases was also found at gene levels. By attributing the assembled transcript to the annotated protein with the highest score, *P. lanuginosus* transcripts had homology with 26,006 unique proteins in RefSeq nr, 24,027 for UniRef90 and 24,178 for UniProtKB TrEMBL database (lower panel of Fig 2A). For *P. maculosus* transcripts, homology was found for 21,228, 19,574 and 19,721 unique proteins in NCBI nr, UniRef90 and UniProtKB TrEMBL, respectively. Majority of the amino acid sequences with homology had very high coverage with

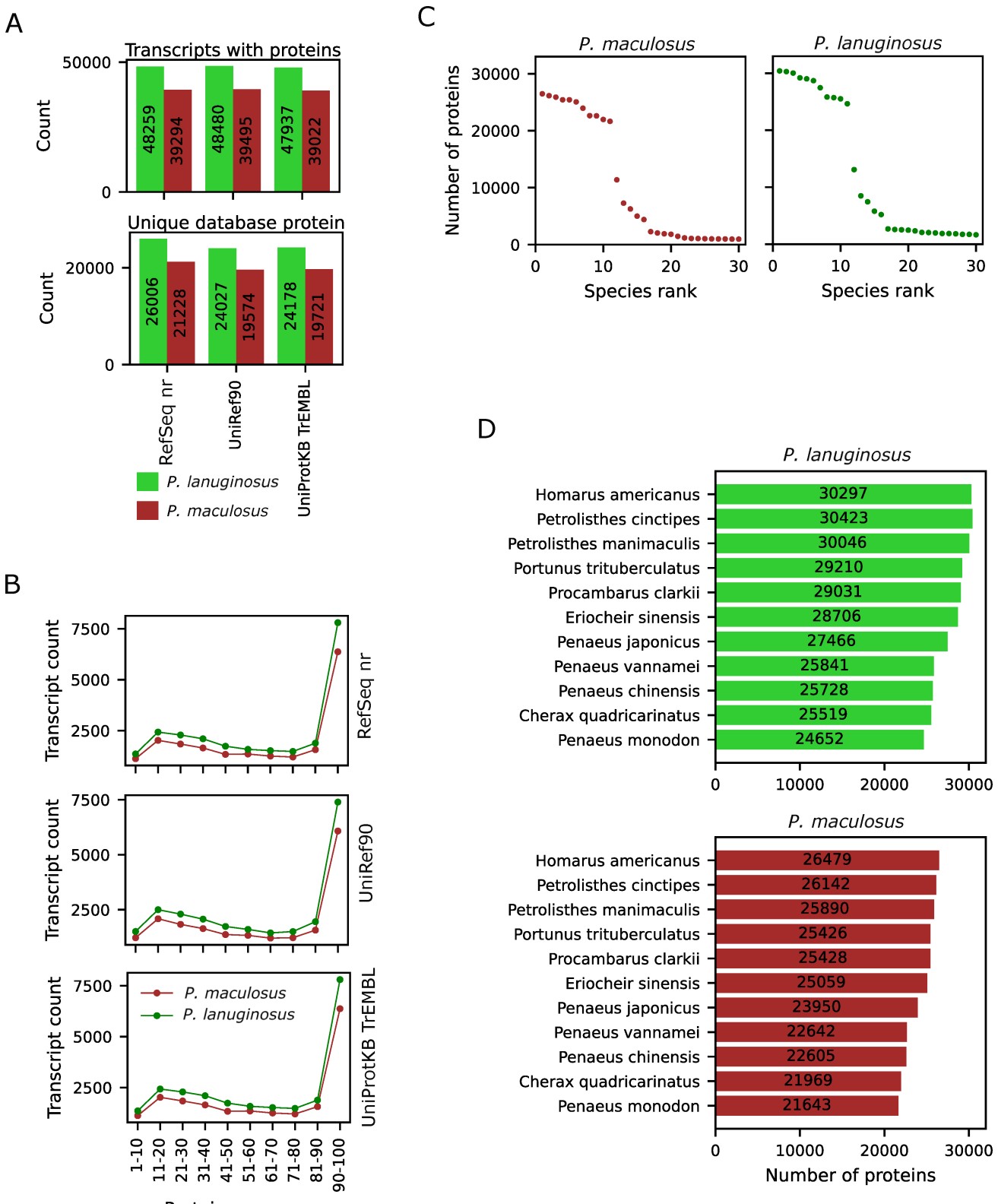

**Fig 2. Annotation of the assembled transcripts with predicted coding abilities.** A. The number of assembled transcripts (upper panel) with significant homologs in the listed database and the corresponding number of unique homologous proteins in the database (lower panel). B. Coverage of the annotated proteins in the assembled transcripts. C. The number of transcripts found in the top 30 represented species. D. The list of the top annotated species with at least 20,000 homologs in the assembled transcripts.

the annotated proteins in the database (Fig 2B). These data showed that a substantial number of predicted amino acid sequences had homology with known proteins.

In order to more efficiently annotate the assembled transcripts, we decided to identify the phylogenetically closely related well-annotated species that could potentially be used as the reference. To identify the closest species with known amino acid sequences, we identified the top 30 species with homologous proteins in both *P. lanuginosus* and *P. maculosus.* Eleven species each had homology to more than 20,000 predicted *P. lanuginosus* and *P. maculosus* transcripts (Fig 2C). Interestingly, the same species were the top closest known species to both *P. lanuginosus* and *P. maculosus.* Based on the number of homologous amino acid sequences, the closest two species were American lobster, *Homarus americanus* and Porcelain crab, *Petrolisthes cinctipes.* The two species probably diverged from *Pagurus sp.* more than 200 million years ago (MYA) [54]. Therefore, the species with the highest numbers of homologous proteins do not share recent evolutionary histories with *Pagurus* species. These results highlight the need for concentrated efforts on the annotation of the transcripts of *Pagurus* sp.

## Phylogenetic analyses and divergence time estimates

The divergence times between two *Pagurus* species and the closest crustacean species suggests that the closest species have not been sequenced or annotated to be included in the databases. We therefore, retrieved the raw reads of multiple crustacean species published as part of the crustacean annotated transcriptome (CAT) database [23]. Using the same pipeline, we assembled the transcripts and predicted amino acids sequences. To have a uniform processing, we also retrieved the short-read sequences of the top 11 species from the database (Figs 2B and 2C) and perform *de novo* transcriptome assembly. In total, 20 species had proteins with homology to more than 20,000 *Pagurus* transcripts. We then extracted transcripts with one-to-one homology between *P. maculosus* and each of the 21 species, including *P. lanuginosus* and the 20 other species. The transcripts were aligned, concatenated and used for phylogenetic analyses. To ensure that the same sites were used in all the species, positions with alignment gaps in any species were completely deleted. A total of 2,035 one-to-one transcripts covering 147,186 gapless amino acids were extracted. These amino acid sequences were used for phylogenetic analyses.

The phylogenetic analyses revealed that *P. maculosus* and *P. lanuginosus* were sister species (Fig 3), consistent with the previous results with the mitochondrial genomes [9,21]. We also found that the next closest species to *P. maculosus* and *P. lanuginosus* was a red king crab, *Paralithodes camtschaticus* which was found to be more closely related than another *Pagurus* species, *Pagurus longicarpus.* This result further confirmed the previous studies reporting "hermit-to-king" hypothesis that *Paralithodes* species evolved from *Pagurus* lineage [55–58]. Importantly, we confirmed that the closest species to the *P. maculosus* and *P. lanuginosus* are not well annotated in the public amino acid databases, highlighting the gap being filled by this study.

Next, we computed the divergence times involving 22 species using MCMC tree. Nine calibration points (S3 Table) were used. Our analysis showed that *P. maculosus* and *P. lanuginosus* diverged around 6.24 (4.43–8.16 95% confidence interval) MYA (Fig 3). The two species diverged from the red king crab, *Par. camtschaticus* around 47.63 (36.14–59.77 95% confidence interval) MYA. The closest identified species pair was two porcelain crabs, *Pet. cinctipes* and *Pet. manimaculis* which were found to have diverged less than 1 MYA. These two *Petrolisthes* species were estimated to have diverged from *Pagurus* species 249.86 (227.16–272.02 95% confidence interval) MYA. Even more diverged was the American lobster, *H. americanus* which was found to have diverged from *Pagurus* species more than 350 MYA. These results showed that *P. maculosus* and *P. lanuginosus* are sister species, and highlight the phylogenetic relationships between *Pagurus* species and selected crustacean species with transcriptome data.

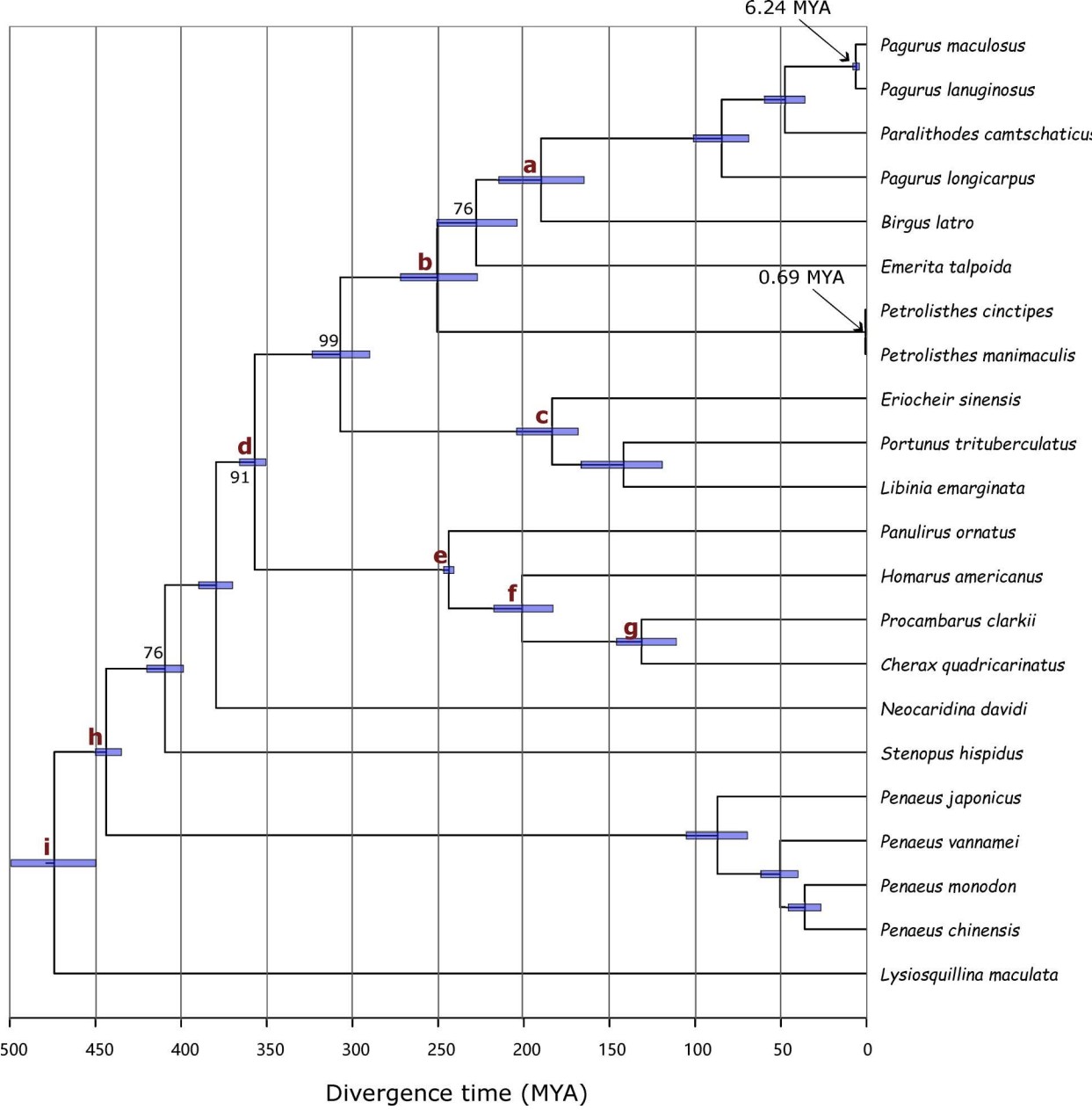

**Fig 3. Phylogenetic analyses and divergence times estimates of the investigated species.** The initial phylogenetic tree was made using maximum likelihood method with 1000 bootstraps. The tree was made using 147,186 gapless amino acid positions from 2,035 aligned one-to-one transcripts. The numbers on the nodes represent the bootstrap values in percentage. Nodes with no indicated bootstrap values had 100% bootstrap support. The calibration points are marked in alphabets a-i, and are described in S3 Table. The divergence times were estimated using MCMCtree.

## Selection pressures after *P. maculosus* and *P. lanuginosus* divergence

After confirming that both *Pagurus* species were sister species, we proceeded to investigate the evolutionary pressures that have shaped the evolution of the two species after their divergence. We first computed the synonymous (dS) and nonsynomous (dN) substitutions between the two species using 22,516 transcripts with a minimum gapless length of 50

amino acids. The estimations of dS and dN from both Yang-Nielsen [42] and Nei-Gojobori [41] methods were very consistent with very high correlation coefficients (Fig 4A). The median dS and dN values from Nei-Gojobori method were 0.0284 and 0.0062, respectively while the vales from Yang- Nielsen method were 0.0246 and 0.0066, respectively (Fig 4B). These results showed that both methods produced similar results.

Having established the consistencies of the two methods, we decided to use Yamg-Nielsen method for the subsequent analyses because the method provides error estimate for each value. We computed the dN/dS ratio to identify the evolutionary pressures acting on each predicted protein-coding transcript. Evolutionary selections of the transcripts were classified into neutral, negative and positive based on the value and error estimates of dS and dN (see Materials and Methods section for details). For neutral transcripts, the estimates of dN and dS and the corresponding errors overlapped. Positively selected transcripts had higher dN values (including the error estimates) while transcripts under purifying selection had higher dS values (including the error estimates). Although higher dN than dS values could also result from relaxed purifying selection [59,60], we simply referred to transcripts with significantly higher dN than dS values as positively selected. The distribution of the dN/dS ratio revealed that dN/dS ratio was less than one for most of the transcripts. (Fig 4C). Because of the inclusion of error estimates in classifying the transcripts into different categories, transcripts from different categories may have similar dN/dS values. Interestingly, dN/dS values for some negatively and positively selected transcripts overlapped the values for some neutrally evolving transcripts while other evolutionarily selected transcripts had substantially higher values, possibly reflecting two distinct levels of selection as presented in Ohta's nearly neutral theory of molecular evolution [61]. The distribution of selection pressures showed that about 55% of the transcripts were under purifying selection while only 5% were under positive selection (Fig 4D), suggesting that purifying selection dominates the evolutionary pressures shaping the gene evolution.

The two species were initially thought to be color morphs of the same species. We therefore checked if the pigmentation genes were positively selected. We first identified assembled *Pagurus* genes that were homologous to known pigmentation genes (S4 Table). Indeed, we found 52 pigmentation genes to be positively selected (Fig 4E). However, the 52 genes only represented about 1% of total one-to-one pigment-related genes (Figs 4E and 4F). Specifically, while 4,260 one-to-one pigmentation genes were found (Fig 4F), only 52 one-to-one pigmentation genes were found to be positively selected (Figs 4E and 4G). The GO enrichment analyses showed that the most significantly enriched term in the positively selected genes was extracellular region (GO:0.0005576) (Fig 4H). These results revealed the dynamics of evolutionary selection after the divergence of *P. maculosus* and *P. lanuginosus.*

## Evolution of gene expressions in *P. maculosus* and *P. lanuginosus*

To investigate the evolution of expression levels in the two species, we took advantage of the high sequence similarity of the two species. It is important to note that the median synonymous divergence was lower than 3%, and the non-synonymous divergence was even lower (0.6%) (Fig 4B). We wanted to see if the assembled transcripts from a species could be used as reference for the quantification of the tissues from the other species. We first asked if the proportion of the homologous genes in the transcriptome of both species were substantial enough. Interestingly, close to 100% of the transcriptomes in both species came from homologous genes (Fig 5A). However, the percentages of transcriptomes with one-to-one homology were lower, with the value dropping to 70% and 79% in a *P. maculosus* leg and head, respectively. This suggests that gene duplications and/or alternative splicing events might have shaped gene evolutions in these two species. Furthermore, the transcriptome percentages when transcripts were considered were even lower, reiterating the potential contribution of splicing to the evolution of gene expression in the two species.

Having established the high sequence similarity and percentages of transcriptomes in the two species, we performed expression quantification again using *P. lanuginosus* reference. We clustered the samples using principal component analyses and heatmaps to identify the two most consistent samples. Using the average expression of the most consistent duplicates, we found that the expression correlation in the head (Pearson's correlation coefficient = 0.76) of the two

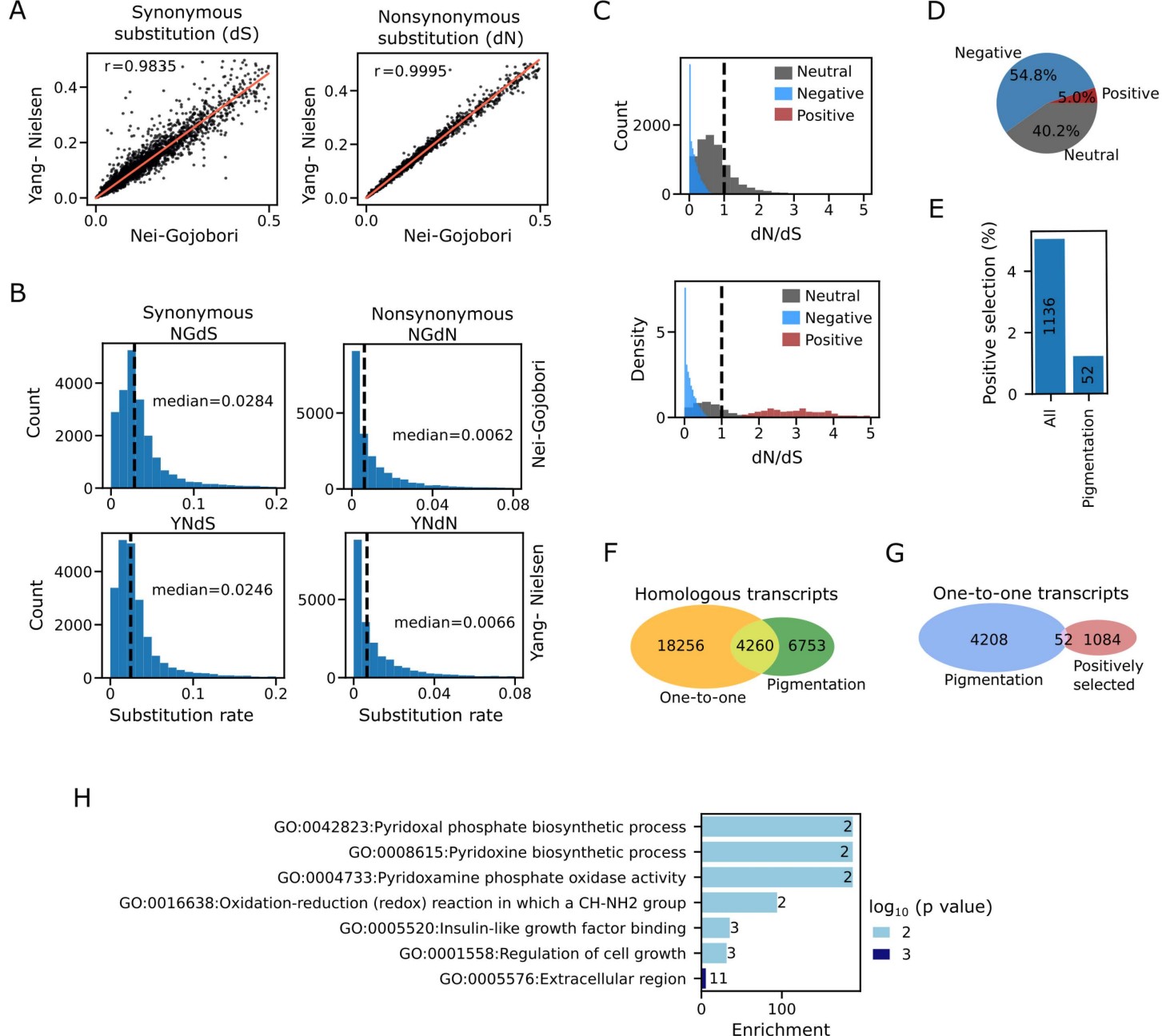

**Fig 4. Evolutionary analyses of the assembled transcripts with protein-coding potentials.** A. Scatter plots showing the relationships between Yang-Nielsen and Nei-Gojobori methods for synonymous (left) and nonsynonymous (right) substitutions. r = Pearson's correlation coefficient. B. The distributions of synonymous and nonsynonymous substitutions computed with Yang-Nielsen and Nei-Gojobori methods. C. The distributions of dN/dS computed from Yang-Nielsen method. The transcripts were grouped into neutral, negative and positive based on the values and error estimates of the dN and dS. D. Pie charts showing the distributions of the transcripts based on the selection pressures. E. The percentages of positively selected transcripts with homology to pigmentation genes. F. Venn diagram showing the overlap between one-to-one and pigmentation genes. G. Venn diagram showing the positively selected pigmentation genes. H. Gene ontology enrichments for positively selected transcripts.

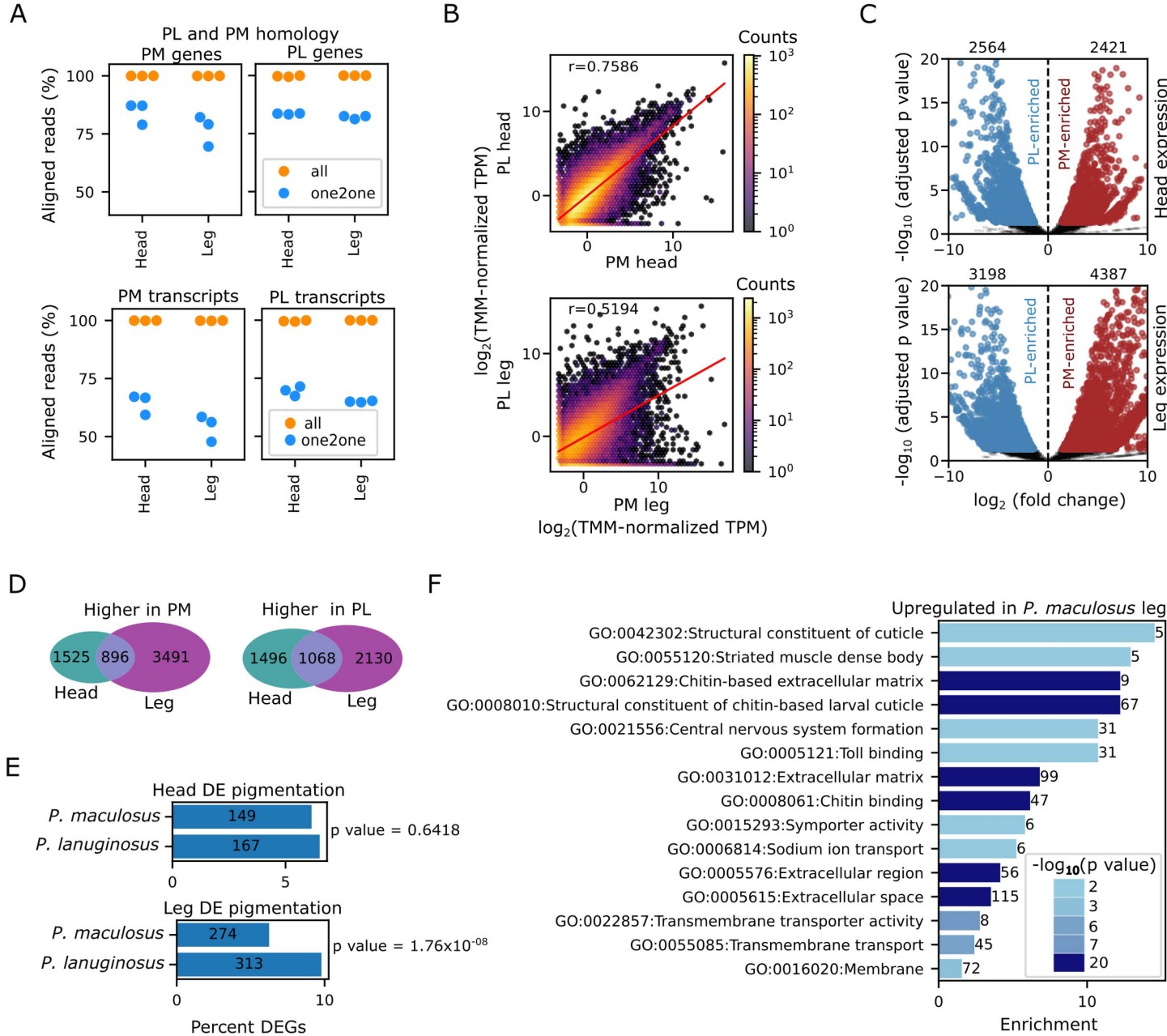

**Fig 5. *Pagurus* gene expression evolution.** A. The percentages of *P. maculosus* and *P. lanuginosus* reads that can be mapped to the genes of other species. Close to 100% percent of the reads could be mapped to the transcripts of the other species. However, when genes/transcripts with one-to-one relationships were used, the percentages dropped. **B**. 2-D hexagonal binning plots showing the relationships between gene expression patterns of the *P. maculosus* (PM) and *P. lanuginosus* (PL) in the head (upper panel) and the leg (lower panel). r = Pearson's correlation coefficient while the red lines represent the linear regression lines. Log$_2$-transformed TMM-normalized expressions were used for the plots C. Volcano plots showing differential gene expression analyses between *P. maculosus* (PM) and *P. lanuginosus* (PL) in the head (upper) and leg (lower). D. Venn diagram showing genes that were consistently upregulated in *P. maculosus* (PM, left panel) and *P. lanuginosus* (PL, right panel) head and leg. E. Percentages of pigmentation genes in head (upper) and leg (lower) differentially expressed genes. P values were computed using Fisher exact tests. F. Gene ontology enrichment analyses for genes upregulated in *P. maculosus* leg. The bar color corresponded to the Bonferroni-corrected hypergeometry p value.

species was higher than the leg correlation (Pearson's correlation coefficient = 0.52), suggesting higher expression evolution in the genes expressed in the leg (Fig 5B). When only the predicted protein-coding genes were used for the correlations (S1A Fig), the correlations became stronger (Pearson's correlation coefficients were 0.84 and 0.60, respectively for head and leg), suggesting divergence expression in the legs across the two species. Interestingly, the trend of higher head correlation persisted when only positively selected genes were used (S1B Fig). Differential gene expression analyses further showed more differentially expressed genes in the leg, compared to the head (Fig 5C), further confirming more divergent expression in the leg and conserved expression in the head across these two species. While 4,985 differentially expressed genes (DEGs) were found in the head, 7,585 DEGs were found in the leg. Also, the patterns of head DEGs were similar across the two species (Fisher exact p value = 0.155) while more genes were upregulated in *P. maculosus* leg (Fisher exact p value = $3.66 \times 10^{-22}$) (Figs 5C and 5D), suggesting tissue-specific evolution of gene expression patterns.

We next asked if the expressions of the same set of genes evolved in both head and leg. Log-fold changes for differentially expressed genes in leg and head were correlated (Spearman's rank coefficient = 0.54) (S1C Fig). For genes that were significantly differentially expressed in both leg and head, the patterns of differential expression were mostly consistent. For genes with higher expression in *P. maculosus,* 896 were found to be common to both head and leg differential expression (Fig 5D). However, a larger proportion of the genes were found to be specific to head or leg comparison. Similar patterns were found for genes with higher expression in *P. lanuginosus*, suggesting that *Pagurus* gene expression evolution involved both shared and tissue-specific components.

To identify the potential functions of the differentially expressed genes, we checked the enrichments of pigmentation genes. Head DEGs included 149 *P. maculosus*-enriched pigmentation genes and 167 *P. lanuginosus*-enriched pigmentation genes (Fig 5E). However, there was no significant bias in pigmentation genes differentially expressed in the head across the two species (p value = 0.64). More pigmentation genes were found in the leg differentially expressed genes. Importantly, more pigmentation genes were found to have higher expression in *P. lanuginosus* leg (p value = $1.76 \times 10^{-08}$). We checked the enrichment of positively selected genes in deferentially expressed gene sets but found no bias in either of the species (S1D Fig), suggesting that positive selection on gene expression was not specifically biased to either species.

Finally, we performed gene ontology enrichments for the differentially expressed genes. As the assembled genes have not been annotated, we first performed GO annotation of the genes using InterProScan 5 to predict the likely gene functions and the potential GO terms for the assembled protein-coding transcripts. Of the 55,714 *P. lanuginosus* with Inter-ProScan annotation, 29,186 could be mapped to at least one GO term (S5 Table). For *P. maculosus*, 23,703 of the 45,038 annotated transcripts could be mapped to at least one GO term. In total, 5,916 (*P. lanuginosus*) and 5,326 (*P. maculosus*) GO terms were found.

GO enrichment analyses showed that the largest enrichments were found in the genes that were upregulated in *P. maculosus* leg (Fig 5F). Gene involved in extracellular activities were significantly enriched. Additionally, genes involved in the chitin-based larval cuticle and chitin binding were upregulated in *P. maculosus* leg. On the contrary, genes upregulated in *P. lanuginosus* leg were enriched in the nucleus, and in peptidase inhibitor and catalytic activities (S1E Fig). Next, we checked the GO enrichment of *P. maculosus*-enriched head DEGs and found that the genes were enriched in ion exchange (S1F Fig). Finally, we checked the GO enrichment of *P. lanuginosus*-upregulated head genes and found that genes involved in sulfation, sulfotransferase and transmembrane transporter activities were enriched (S1E Fig). These results showed that genes upregulated in different species and different tissues performed different functions. Taken together, the results highlight how gene expressions have evolved after the divergence of *P. maculosus and P. lanuginosus.*

## Discussion

*Pagurus maculosus* and *P. lanuginosus* are two intertidal hermit crab species found in the Japanese water. They were initially thought to be color morphs of the same species. However, subsequent studies demonstrated that they are two

separate species with unique features [7,9–11,21]. In this study, we report the analyses of the head (cephalothorax) and leg (pereopod) transcriptomes of the two species. We performed independent *de novo* transcript assembly for the two species, and confirmed that our transcript assembly did not lead to substantial loss of transcriptome data. Although almost all reads of one species could be mapped to the transcripts of the other species, a sizable proportion of the transcripts did not have one-to-one correspondence between the two species. Also, consideration at gene levels had higher proportion of one-to-one correspondence between the two species. These suggest that gene duplication and alternative splicing have played important roles in shaping the evolution of gene expressions in the two species.

We quantified the genetic distance between the two species. Using Yang-Nielsen method [42], the synonymous and nonsynonymous substitution rates for the two species were estimated to be 2.46% and 0.66%, respectively. These values were lower than 3.4–4.1% reported from nucleotide sequences of selected mitochondrial loci [21]. This discrepancy might be due to the differences in the evolutionary rates of mitochondrial and nuclear genes [62]. In addition, while the mitochondrial value was computed using Kimura 2-parameter on mitochondrial regions without segregation into synonymous and nonsynonymous rates, the evolutionary distances computed in this study were computed from predicted protein-coding transcripts. Based on synonymous and nonsynonymous substitutions, majority of the genes were found to undergo purifying selection, and only 5% were found to be undergoing positive selection. Interestingly, we found that substantial number of genes under selection are undergoing slightly neutral evolution [61,63]. This observation is interesting as the two species are sometimes found in the same ecosystem, suggesting that the two species might not have been subjected to drastically different pressures. The role of geographical isolation in shaping speciation and molecular evolution after speciation would be an interesting future study.

The search of public databases revealed that the closest annotated species to *P. lanuginosus* was the porcelain crab, *Pet. cinctipes,* which was estimated to have diverged around 250 MYA. However, our analyses revealed that the pair of *P. lanuginosus* and *P. maculosus* diverged from *Par. camtschaticus* about 36.1–59.8 MYA. These suggest that species closer to *Pagurus* have not been well annotated or their annotations have not been integrated into public protein databases. Supporting the later possibility, indeed transcriptome data of a number of species have recently been published [23,64–66]. Using the transcriptome of multiple related species, we estimated the divergence time of *P. lanuginosus* and *P. maculosus* to be 6.24 MYA (4.43–8.16; 95% confidence interval). This divergence time gives the rate of synonymous substitutions of $3.9 \times 10^{-9}$, a rate similar to what was found in mammals [46]. Consistent with the previous reports that genus *Pagurus* is not monophyletic [4,9,67,68], we showed that the red king crab, *Par. camtschaticus* is more closely related to *P. maculosus* than *P. longicarpus.*

Gene expression evolution was found to be different across tissues with the head having higher correlation, suggesting slower gene expression evolution in the head when compared to the leg. This observation was similar to mammalian studies that investigated conserved noncoding sequences to infer that gene expression evolution varied across tissues [69–71]. Interestingly, we found that gene expression evolution was not symmetric. Genes involved in pigmentation were found to be significantly upregulated in *P. lanuginosus.* We found that genes upregulated in *P. maculosus* leg were enriched in extracellular space, larval cuticle, chitin binding and extracellular matrix, suggesting relationships between the gene expression changes and previously reported morphological changes [7,10,11]. Taken together, this study uncovers the molecular changes that have ensued the divergence of *P. maculosus* and *P. lanuginosus* and the potential relationships with the reported morphological changes.

## Supporting information

**S1 Fig. Gene expression evolution and the nature of genes enriched across the species.** A. 2-D hexagonal binning plots showing the relationships between protein-coding gene expression patterns of the *P. maculosus* (PM) and *P. lanuginosus* (PL) in the head (upper panel) and the leg (lower panel). B. 2-D hexagonal binning plots showing the relationships

between the positively selected protein-coding gene expression patterns of the *P. maculosus* (PM) and *P. lanuginosus* (PL) in the head (upper panel) and the leg (lower panel). For A and B, r = Pearson's correlation coefficient while the red lines represent the linear regression lines. Log$_2$-transfromed TMM-normalized expressions were used for the plots C. Scattered plot showing the relationship between the log-fold changes of genes differentially expressed in head and/or leg. Each dot represents a gene. Negative values represent *P. lanuginosus* enrichment while positive values represent *P. maculosus* enrichment. Rho = Spearman'r correlation coefficient. D. Species bias of positively selected differentially expressed head (upper) and leg (lower) genes. P values were computed using Fisher exact tests. Gene ontology enrichments of genes are upregulated in *P. lanuginosus* leg (E), *P. maculosus* head (F) and *P. lanuginosus* head (G). The bar color corresponded to the Bonferroni-corrected hypergeometry p value.
(TIF)

**S1 Table. Details of Transcriptome sequencing.**
(XLSX)

**S2 Table. Number of homologous transcripts and genes in different databases.**
(XLSX)

**S3 Table. Calibrations used for divergence time estimates.**
(XLSX)

**S4 Table. The list of known pigmentation genes.**
(XLSX)

**S5 Table. InterproScan annotation statistics for the assembled transcripts.**
(XLSX)

## Acknowledgments

The authors are thankful to Mr. Nakano, R., Ms. Nakano, M., and the members of SMBL for their help in collecting the study animals.

## Author contributions

**Conceptualization:** Zakea Sultana, Masafumi Nozawa, Kazuho Ikeo, Akira Asakura, Tomoyuki Nakano.

**Data curation:** Zakea Sultana, Isaac Adeyemi Babarinde, Kazuho Ikeo, Tomoyuki Nakano.

**Formal analysis:** Isaac Adeyemi Babarinde.

**Funding acquisition:** Tomoyuki Nakano.

**Investigation:** Zakea Sultana, Isaac Adeyemi Babarinde, Masafumi Nozawa, Akira Asakura.

**Methodology:** Zakea Sultana, Isaac Adeyemi Babarinde, Masafumi Nozawa, Kazuho Ikeo, Akira Asakura, Tomoyuki Nakano.

**Project administration:** Tomoyuki Nakano.

**Resources:** Isaac Adeyemi Babarinde, Kazuho Ikeo, Akira Asakura, Tomoyuki Nakano.

**Software:** Isaac Adeyemi Babarinde.

**Supervision:** Kazuho Ikeo, Akira Asakura.

**Validation:** Zakea Sultana, Isaac Adeyemi Babarinde, Masafumi Nozawa.

**Visualization:** Isaac Adeyemi Babarinde.

**Writing – original draft:** Zakea Sultana, Isaac Adeyemi Babarinde.

**Writing – review & editing:** Zakea Sultana, Isaac Adeyemi Babarinde, Masafumi Nozawa, Kazuho Ikeo, Akira Asakura, Tomoyuki Nakano.

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
