## [Decision Letter · Decision Letter 0]

27 Jun 2025

Dear Dr. Babarinde,

Thank you for submitting your manuscript to PLOS ONE. After careful consideration, we feel that it has merit but does not fully meet PLOS ONE’s publication criteria as it currently stands. Therefore, we invite you to submit a revised version of the manuscript that addresses the points raised during the review process.

We look forward to receiving your revised manuscript.

Kind regards,

Feng ZHANG, Ph.D.

Academic Editor

PLOS ONE

Journal Requirements:

“This study was supported by the funding of Research Organization of Information and Systems (ROIS) for special collaboration research student program of National Institute of Genetics (NIG) and Kyoto University, Japan”

“The authors are thankful to Mr. Nakano, R., Ms. Nakano, M., and the members of SMBL for their help in collecting the study animals. This study was supported by the funding of Research Organization of Information and Systems (ROIS) for special collaboration research student program of National Institute of Genetics (NIG) and Kyoto University, Japan.”

“This study was supported by the funding of Research Organization of Information and Systems (ROIS) for special collaboration research student program of National Institute of Genetics (NIG) and Kyoto University, Japan”

7. PLOS requires an ORCID iD for the corresponding author in Editorial Manager on papers submitted after December 6th, 2016. Please ensure that you have an ORCID iD and that it is validated in Editorial Manager. To do this, go to ‘Update my Information’ (in the upper left-hand corner of the main menu), and click on the Fetch/Validate link next to the ORCID field. This will take you to the ORCID site and allow you to create a new iD or authenticate a pre-existing iD in Editorial Manager.

8. We are unable to open your Supporting Information file [HermitCrab_SupplementaryTables.xlsx and S1Fig.eps]. Please kindly revise as necessary and re-upload.

Reviewers' comments:

Reviewer's Responses to Questions

**Comments to the Author**

1. Is the manuscript technically sound, and do the data support the conclusions?

Reviewer #1: Yes

Reviewer #2: Yes

2. Has the statistical analysis been performed appropriately and rigorously?

Reviewer #1: Yes

Reviewer #2: Yes

3. Have the authors made all data underlying the findings in their manuscript fully available?

Reviewer #1: Yes

Reviewer #2: Yes

4. Is the manuscript presented in an intelligible fashion and written in standard English?

Reviewer #1: No

Reviewer #2: Yes

Reviewer #1: I believe that this paper is valuable for publication. However, as mentioned below, it needs some revisions.

L.38, 39: leg of xxx leg?

L.40: chin-based?

L.52: what is “those species”?

L.52: “Sibling species and cryptic species are sometimes not properly identified”. Because taxa not known are cryptic species.

L.93: “RNA samples were extracted separately from the head and leg” Both parts contain muscle, membrane, and exoskeleton or something else, but RNA was extracted from muscle. First, the material must be properly identified.

BLAST not blast

L.211-212: This must be placed in Materials and Methods.

L.215: remove “for each sample” How many individuals per species were used?

L215-216: This must be placed in Materials and Methods.

L.219-221: This must be placed in Materials and Methods.

L.242-244: This must be placed in Materials and Methods.

L.321-324: References 9 and 18 addressed only the genus Pagurus. Therefore, if the authors are referring to the non-monophyletic status of Pagurus in relation to the family Lithodidae, they should cite other references (e.g., Cunningham et al., 1992; Zaklin, 2001; Morrison et al., 2002; Blacken-Grissom et al., 2013).

L.505: respectively

The generic abbreviations for Pagurus, Paralithodes, and Petrolisthes should be clear. For example, they could be Pag., Par., and Pet.

L.505-: I guess that almost all nucleotide substitutions in protein-coding mitochondrial DNA between the two Pagurus species are synonymous.

L.519: “found” should be “estimated”

L.525: specify the two species.

L.529: A recent study on ITS1 sequence divergence and length (Chow et al. 2023, Crust. Res. 52) showed well separate status between Pagurus and Lithodid. No discussion for this?

Reviewer #2: The MS investigates the molecular and gene expression evolution of two closely related Pagurus species using transcriptome sequencing. The results sheds light on the evolutionary history of the two species at the molecular level. This MS demonstrates substantive content, sound data analysis, and reliable results, presented through well-structured writing that meets the criteria for publication readiness.

**Do you want your identity to be public for this peer review?** For information about this choice, including consent withdrawal, please see our Privacy Policy

Reviewer #1: No

Reviewer #2: No

---

## [Author Response · Author response to Decision Letter 1]

25 Jul 2025

We appreciate the editor and the reviewers for the time and efforts spent in reviewing our manuscript. We are particularly grateful for the suggestions towards the improvement of the manuscript. We have meticulously revised the manuscript based on the suggestions and comments from the editor and the reviewers. We have also corrected other typographical or grammatical errors that we encountered. All changes have been marked in one of the uuploaded documents. Specific response and changes made in response to each comment are itemized below (in blue font).

Response to the editor

We are grateful to the editor for the comments. We have responded to each raised comment with blue font below.

We appreciate the editor for pointing this out. The first page has been revised to be consistent with PLOS ONE’s style. We have also revised the listing of supplementary figures and tables to be consistent with the PLOS ONE’s style.

We have updated the Materials and Methods section by adding the following sentences. “Sample collection permit for experimental research with number 5-2 was granted by Wakayama Prefecture, Japan. Researches on Pagurus species do not require any special approval. All experiments complied with institutional regulations and Japanese policy on animal use [22].” We also corrected the sampling date to reflect the actual date when the specific sequenced samples were collected.

The grants used for the study do not have any special grant number as they are mostly in-house grants.

“This study was supported by the funding of Research Organization of Information and Systems (ROIS) for special collaboration research student program of National Institute of Genetics (NIG) and Kyoto University, Japan”

We declare that "The funders had no role in study design, data collection and analysis, decision to publish, or preparation of the manuscript."

“The authors are thankful to Mr. Nakano, R., Ms. Nakano, M., and the members of SMBL for their help in collecting the study animals. This study was supported by the funding of Research Organization of Information and Systems (ROIS) for special collaboration research student program of National Institute of Genetics (NIG) and Kyoto University, Japan.”

“This study was supported by the funding of Research Organization of Information and Systems (ROIS) for special collaboration research student program of National Institute of Genetics (NIG) and Kyoto University, Japan”

As suggested by the editor, we have deleted the funding-related text from the manuscript. And we would like to leave the funding statement as it is, that is, “This study was supported by the funding of Research Organization of Information and Systems (ROIS) for special collaboration research student program of National Institute of Genetics (NIG) and Kyoto University, Japan”

We have made the raw read data publicly available (as from July 22nd, 2025) using the new accessions provided in the manuscript. The data can now be assessed from https://www.ncbi.nlm.nih.gov/bioproject/PRJDB35803. We would like to point out that we discovered that the biological replicates were not correctly entered in the original submission (https://www.ncbi.nlm.nih.gov/bioproject/?term=PRJDB7354). That is why we had to revise the submission and now have new accessions for the project and the samples. The manuscript has been appropriately revised to state “All raw reads generated in this study are available in the DDBJ Sequence Read Archive (DRA) under the BioProject accession number PRJDB35803. The data sets supporting the results presented here are available with the accession numbers DRR708008-DRR708019.”

7. PLOS requires an ORCID iD for the corresponding author in Editorial Manager on papers submitted after December 6th, 2016. Please ensure that you have an ORCID iD and that it is validated in Editorial Manager. To do this, go to ‘Update my Information’ (in the upper left-hand corner of the main menu), and click on the Fetch/Validate link next to the ORCID field. This will take you to the ORCID site and allow you to create a new iD or authenticate a pre-existing iD in Editorial Manager.

ORCID ID has been linked and validated in Editorial Manager.

8. We are unable to open your Supporting Information file [HermitCrab_SupplementaryTables.xlsx and S1Fig.eps]. Please kindly revise as necessary and re-upload.

We have confirmed that the files are still in good shape. We suspect that the problem might have arisen during file uploading. We will re-upload the files during the submission of the revised manuscript. To ensure that the files can be opened, supplementary figure will be saved in two formats (.eps and .tiff). An additional compressed file of the supplementary figure and table files will also be attached. Multiple uploading of the same files is to ensure the ability of the files to be opened.

The revised manuscript now included references 18-20 supporting the growing use of molecular barcodes. We have listed reference 22 about institutional regulations and Japanese policy on animal use. In addition, we have cited new papers (references 55-58) suggested by a reviewer.

Response to reviewers

Reviewer #1: I believe that this paper is valuable for publication. However, as mentioned below, it needs some revisions.

We appreciate the reviewer for the time and efforts towards the improvement of our manuscripts. We are sincerely grateful for the multiple corrections that the reviewer has suggested. We have incorporated the reviewer’s suggestions and corrections in the revised version of the manuscript. We have also listed point-by-point response to the reviewer’s suggestion and comments.

L.38, 39: leg of xxx leg?

This has been revised by deleting “leg”.

L.40: chin-based?

“chin-based” has been changed to “chitin-based”

L.52: what is “those species”?

“those species” has been changed to “Pagurus species”

L.52: “Sibling species and cryptic species are sometimes not properly identified”. Because taxa not known are cryptic species.

This sentence has been changed to “Consequently, many sibling species remain cryptic species that are difficult to recognize using traditional classic methods”

L.93: “RNA samples were extracted separately from the head and leg” Both parts contain muscle, membrane, and exoskeleton or something else, but RNA was extracted from muscle. First, the material must be properly identified.

We apologize for this confusion. RNA extraction was actually done using the whole leg or head as pointed out in Figures 1A and B, and not just for the muscle. We have replaced “muscle” with “head or leg” in the Methods section.

BLAST not blast

All instances of “blast” to “BLAST”, including “BLASTN” and “BLASTP”.

L.211-212: This must be placed in Materials and Methods.

This has been moved.

L.215: remove “for each sample” How many individuals per species were used?

This has been revised. Three individuals were used per species.

L215-216: This must be placed in Materials and Methods.

This has been removed.

L.219-221: This must be placed in Materials and Methods.

This has been deleted. We have revised the following sentence to read “Removal of potentially noisy transcripts produced 203,793 transcripts, including 140,954 genes for P. maculosus, and 259,375 transcripts, including 179,685 genes for P. lanuginosus (Table 1)”

L.242-244: This must be placed in Materials and Methods.

This has been deleted. We have added the following to give the background to the result: “Transcriptome assembly has the potential to retrieve both coding and noncoding transcripts. We therefore checked the proportion of both coding and noncoding transcripts in our transcriptome assembly (see Materials and Methods)”

L.321-324: References 9 and 18 addressed only the genus Pagurus. Therefore, if the authors are referring to the non-monophyletic status of Pagurus in relation to the family Lithodidae, they should cite other references (e.g., Cunningham et al., 1992; Zaklin, 2001; Morrison et al., 2002; Blacken-Grissom et al., 2013).

We appreciate the reviewer. We have rephrased the sentence to focus on the evolution of Paralithodes species from Pagurus lineage. As suggested by the reviewer, we have added Cunningham et al. (1992), Morrison et al. (2002) and Bracken-Grissom et al. (2013), as suggested by the reviewer. We also added Chow et al. (2023) which supports the same hypothesis. The sentence now reads “This result further confirmed the previous studies reporting “hermit-to-king” hypothesis that Paralithodes species evolved from Pagurus lineage [55–58].”.

L.505: respectively

“respective” has been changed to “respectively”

The generic abbreviations for Pagurus, Paralithodes, and Petrolisthes should be clear. For example, they could be Pag., Par., and Pet.

We appreciate the reviewer for this suggestion. Since our focus is on Pagurus, we have retained P. for Pagurus genus while using Par. and Pet. for Paralithodes, and Petrolisthes, respectively, as suggested by the reviewer.

L.505-: I guess that almost all nucleotide substitutions in protein-coding mitochondrial DNA between the two Pagurus species are synonymous.

In Sultana et al., two rRNAs, two ATPs, Cox1, Cox2, Cox3, Nn2, and Nd3 mitochondrial regions were analyzed to compute overall nucleotide distance. Although the mitochondrial genes included those that code and those that do not code for proteins, all regions were analyzed together without considering nucleotide substitution type. Indeed, one of the motivations for this study was to more systematically investigate nucleotide evolution. In this study, we have separately computed synonymous and nonsynonymous substitution rates in larger gene set.

L.519: “found” should be “estimated”

This has been corrected.

L.525: specify the two species.

“the two species” has been changed to “P. lanuginosus and P. maculosus”

L.529: A recent study on ITS1 sequence divergence and length (Chow et al. 2023, Crust. Res. 52) showed well separate status between Pagurus and Lithodid. No discussion for this?

We appreciate the reviewer for calling our attention to this paper. Interestingly, the paper showed that lithodoid species had unique internal transcribed spacer 1 (ITS1), which evolved from those found among paguroid species. Their results support the evolution of monophyletic Lithodid from Pagurus lineage. In other words, their findings support the “hermit-to-king” crab hypothesis, as our findings do. Please see Fig. 8 of Chow et al. (2023) which presents the hypothetical evolutionary relationship among hermit crab families. We have included the reference in our manuscript.

Again, we thank the reviewer for their contribution. We hope all the concerns of the reviewer has been addressed.

Reviewer #2: The MS investigates the molecular and gene expression evolution of two closely related Pagurus species using transcriptome sequencing. The results sheds light on the evolutionary history of the two species at the molecular level. This MS demonstrates substantive content, sound data analysis, and reliable results, presented through well-structured writing that meets the criteria for publication readiness.

We appreciate the reviewer for the positive feedback about our manuscript. Thank you.

---

## [Editor Report · Decision Letter 1]

29 Jul 2025

Transcriptome sequencing reveals the evolutionary histories and gene expression evolution in two related Pagurus species

PONE-D-25-30439R1

Dear Dr. Isaac Adeyemi BABARINDE,

We’re pleased to inform you that your manuscript has been judged scientifically suitable for publication and will be formally accepted for publication once it meets all outstanding technical requirements.

Kind regards,

Feng ZHANG, Ph.D.

Academic Editor

PLOS ONE
---

## [Editor Report · Acceptance letter]

PONE-D-25-30439R1

PLOS ONE

Dear Dr. Babarinde,

I'm pleased to inform you that your manuscript has been deemed suitable for publication in PLOS ONE. Congratulations! Your manuscript is now being handed over to our production team.

Kind regards,

on behalf of

Dr. Feng ZHANG

Academic Editor

PLOS ONE